# Unlocking Video-LLM via Agent-of-Thoughts Distillation

## Abstract

This paper tackles the problem of video question answering (VideoQA), a task that often requires multi-step reasoning and a profound understanding of spatial-temporal dynamics. While large generative video-language models perform well on benchmarks, they often lack explainability and spatial-temporal grounding. In this paper, we propose **A**gent-**o**f-**T**houghts **D**istillation (**AoTD**), a method that enhances generative models by incorporating automatically generated Chain-of-Thoughts (CoTs) into the instruction-tuning process. Specifically, we leverage an agent-based system to decompose complex questions into sub-tasks, and address them with specialized vision models, the intermediate results are then treated as reasoning chains. We also introduce a verification mechanism using a large language model (LLM) to ensure the reliability of generated CoTs. Extensive experiments demonstrate that AoTD improves the performance on multiple-choice and open-ended benchmarks.

## 1 Introduction

Video Question Answering (VideoQA) is a critical task in the computer vision community, offering a natural interface for human-machine interaction through language (Yu et al., 2019; Wu et al., 2021; Xiao et al., 2021; Pătrăucean et al., 2023). This synergy of visual content and language enhances the accessibility of AI systems for the general public, allowing users to query complex visual content with everyday language. By encompassing tasks such as action recognition, object detection, and scene understanding, VideoQA serves as a comprehensive benchmark for evaluating AI's ability to interpret videos, addressing the fundamental questions of 'who,' 'what,' 'when,' and 'where' that are crucial to understand daily life activities, pushing the boundaries of what AI systems can interpret from dynamic visual content.

Recent literature in VideoQA has highlighted two key directions. The first focuses on training large generative video-language models (Video-LLMs) through direct instruction-tuning, where videos are only paired with questions and answers (Alayrac et al., 2022; Lin et al., 2024; Maaz et al., 2024; Cheng et al., 2023). While these models have shown exceptional performance on public benchmarks, they often lack explainability and struggle with spatio-temporal grounding. This limitation hinders their ability to provide clear reasoning, which is essential for real-world applications where transparency and interpretability are critical (Mitra et al., 2023).

In contrast, an emerging approach focuses on agent-based systems (Surís et al., 2023; Gupta & Kembhavi, 2023; Hu et al., 2024b), which break down complex questions into simpler sub-tasks. Each sub-task is handled by specialized tools, and the results are aggregated to generate a final answer. This approach theoretically offers greater interpretability, as the reasoning process is divided into explainable steps that can be independently assessed. However, our experiments indicate that current video understanding tools are not strong enough for building reliable agent-based systems. Additionally, the high memory demands and time-consuming nature of these systems present significant challenges for their practical use.

In this paper, we propose enhancing the capabilities of large generative video-language models by incorporating automatically generated Chain-of-Thoughts (CoTs) into the instruction-tuning process. Our approach draws inspiration from agent-based systems, which break down complex questions into a sequence of sub-tasks, each handled by specialized models (Fan et al., 2024; Mahmood et al., 2024; Min et al., 2024). We use the outputs from these specialized models to construct CoTs that

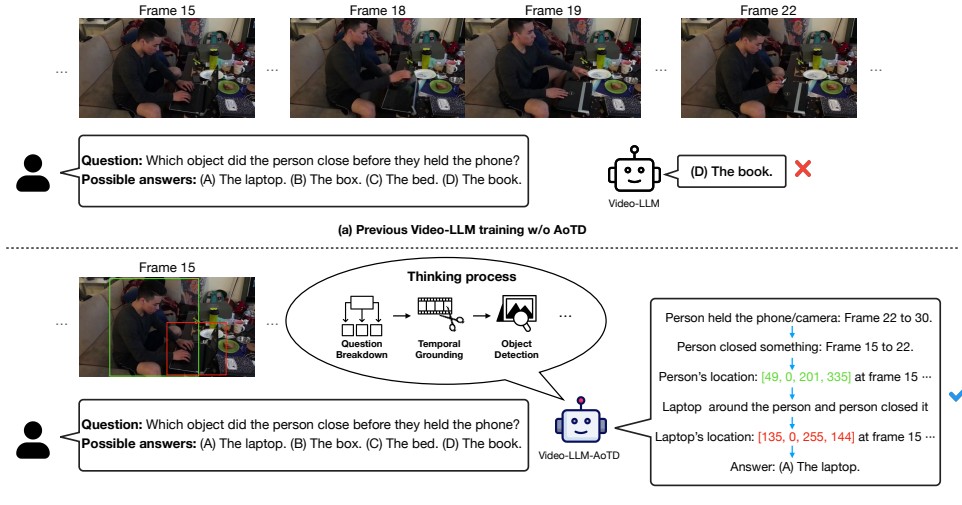

Figure 1: Our method, **AoTD**, distills multi-step reasoning and spatio-temporal understanding into a single generative video-language model. When addressing complex VideoQA tasks, the model trained with AoTD (as shown in (b)) enables to generate a step-by-step reasoning to get the correct answer. In contrast, previous models trained solely on question-answer pairs (as in (a)) generate only a final answer, often without intermediate reasoning, which can lead to incorrect conclusions.

explicitly represent step-by-step reasoning paths, capturing the reasoning processes that generative models typically struggle to model independently.

To ensure the reliability of the constructed CoTs, we systematically evaluate existing models and tools for atomic video understanding tasks, such as action recognition (Weng et al., 2023; Wang et al., 2024) and language grounding (Lin et al., 2023), using a well-annotated dataset. This allows us to identify the best-performing tools for each sub-task, preparing for effective CoTs distillation. This process also serves as an evaluation of the broader capabilities of visual models in more general and complex scenes, offering guidance for future exploration in the computer vision community. Additionally, we introduce a verification mechanism with a large language model (LLM), to assess whether the generated CoTs follow a clear, step-by-step reasoning process and contain useful information for answering the question. This filters out low-quality or logically inconsistent reasoning paths. The verified, high-quality CoTs are then distilled into large generative video-language models, enhancing both performance and the interpretability of their outputs. By combining the strengths of both approaches, our method balances performance with transparency, leading to the development of more robust, accurate, and interpretable VideoQA systems.

In summary, our contributions are three-fold: *First*, we propose a novel approach for enhancing large generative video-language models (Video-LLMs) by distilling high-quality Chain-of-Thoughts (CoTs) into their instruction tuning. These CoTs capture step-by-step reasoning paths, improving both the model's performance and its interpretability; *Second*, to automatically construct the CoTs for any datasets, we employ an agent-based system to decompose complex VideoQA questions into simpler sub-tasks, leveraging off-the-shelf vision models to handle each sub-task. The intermediate outputs from these models can therefore be collected as CoTs for addressing the corresponding visual question; *Third*, we demonstrate through extensive experiments that our distilled model outperforms existing methods across both multiple-choice and open-ended VideoQA benchmarks, enabling to deliver not only accurate answers but also clear and comprehensive reasoning explanations.

## 2 AGENT-OF-THOUGHTS DISTILLATION

In this paper, we propose a novel approach, termed Agent-of-Thought Distillation (AoTD), to enhance the Video-LLMs by training them with multi-step chain-of-thoughts (CoTs). Specifically, we

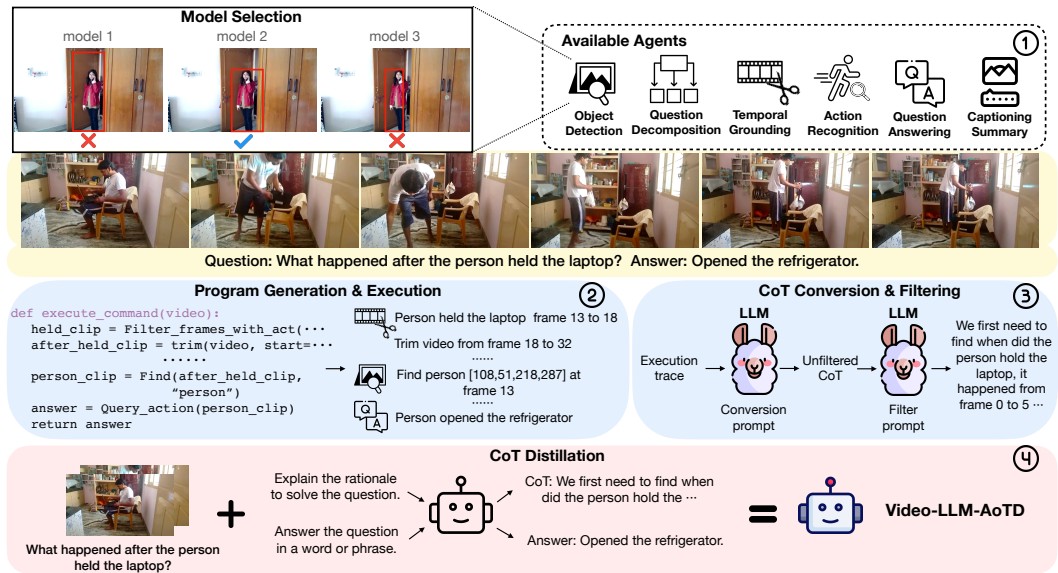

Figure 2: Overview on Agent-of-Thoughts Distillation (AoTD). **Step 1:** Selecting best-performing agents for each sub-task to construct an agent-based system. **Step 2:** Decomposing question into executable program and leveraging chosen models to solve it sequentially to generate execution trace. **Step 3:** The execution trace is converted and filtered by LLM to produce high quality natural language CoTs. **Step 4:** Distilling CoTs into Video-LLM with two forms of prompt, allowing it achieve a balance between concise answers and comprehensive rationales. The final model is Video-LLM-AoTD.

begin by developing an agent-based video understanding system to generate multi-step reasoning chains that address complex video questions. These reasoning chains are then distilled into one Video-LLM through instruction tuning. By combining the strengths of agent-based systems and large generative models, our proposed AoTD enables to build more reliable and interpretable Video Question Answering systems.

## 2.1 PROBLEM FORMULATION

Given a video clip with $t$ frames, $\mathcal{V} = \{x_1, \ldots, x_t\}$, and a set of $n$ questions $\mathcal{Q} = \{q_1, q_2, ..., q_n\}$, our goal is to train a Video-LLM capable of producing both concise answers and comprehensive rationales. Depending on the suffix prompt $p_s$, the model can generate different types of outputs. The process can be formulated as:

$$\{a_i, \mathcal{S}_i\} = \Phi(\mathcal{V}, q_i, p_s), \quad \text{where } \mathcal{S}_i = \{\varnothing\} \text{ or } \{s_{i,1}, s_{i,2}, \ldots, s_{i,k}\}$$

where $q_i$ denotes the $i$-th question, $a_i$ is the answer in free-form text, and $\mathcal{S}_i$ represents the rationale, consisting of the multi-step reasoning process. If the prompt specifies to only generate the answer, $\mathcal{S}_i = \{\varnothing\}$. Otherwise, if the prompt requires the generation of rationales, $\mathcal{S}_i = \{s_{i,1}, s_{i,2}, \ldots, s_{i,k}\}$, where each $s_{i,j}$ corresponds to a reasoning step.

**Discussion.** Unlike existing models that are typically instruction-tuned on VideoQA datasets using simple question-answer pairs, which bypass the intermediate thought process, our approach emphasizes the importance of training with rationales, or chain-of-thoughts (CoTs). In the following section, we outline the process for generating high-quality CoTs from existing VideoQA datasets.

## 2.2 COTS CONSTRUCTION WITH AGENT-BASED SYSTEM

Recent work, such as STAR (Wu et al., 2021), has introduced executable symbolic programs that can directly decompose questions into sub-tasks. When combined with scene graphs that contain comprehensive video information from key frames—such as object locations, interactions, and actions—these programs facilitate the generation of concise Chain-of-Thoughts (CoTs) through the

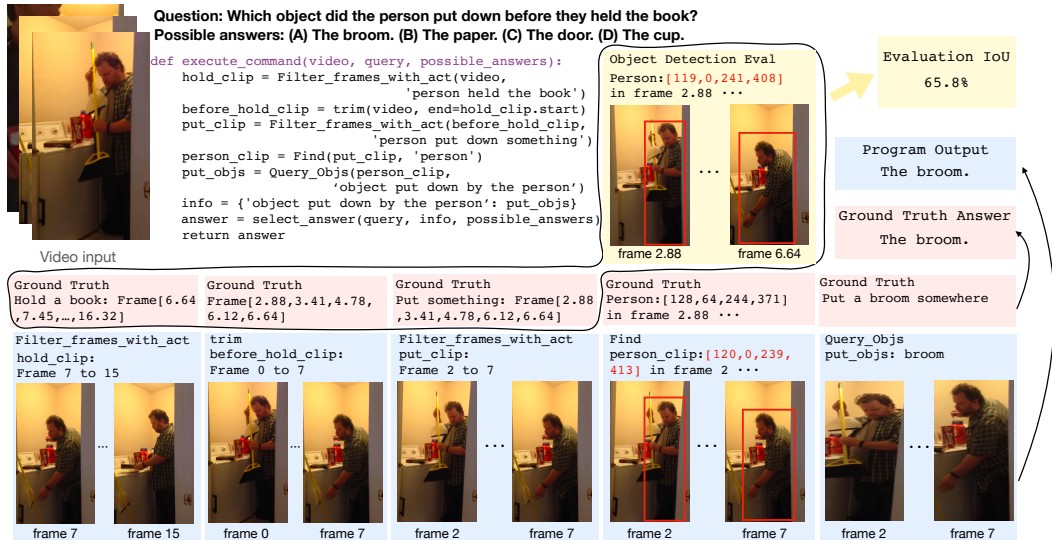

Figure 3: Program execution process in an agent-based system. We uniformly sample 32 frames from the video, and to ensure scale consistency, the frame ids of key frames are normalized into these 32 frames. The blue boxes represent the program execution steps, the red boxes denote the ground truth for each step.

direct execution of symbolic operations. However, datasets of this nature are limited in scale. In response to this limitation, we propose an agent-based system capable of breaking down complex questions into simpler sub-tasks, utilizing off-the-shelf vision models. The intermediate outputs from this system can then be employed to construct CoTs for any existing VideoQA dataset.

**Agent-based VideoQA.** Given a video input ($\mathcal{V}$), questions ($\mathcal{Q}$), and a set of visual models ($\mathcal{M} = \{\phi_{act}, \phi_{det}, \dots, \phi_{qa}\}$), an LLM-based agent core ($\pi(\cdot)$) processes the question along with the documentation of the visual models ($\mathcal{T}$), which includes variables and functionalities. The agent then decomposes the question into sub-tasks and addresses them by invoking the corresponding visual models. It is important to note that the visual models can be arranged in various orders depending on the specific question, ensuring flexibility in problem-solving.

Specifically, in the example illustrated in Fig. 3, the question is first decomposed into a series of sub-tasks, including temporal grounding, object detection, and question answering. The corresponding specialized models are then executed sequentially to address these sub-tasks, ultimately yielding the final answer:

$$\{\phi_{ground}, \phi_{det}, \phi_{qa}\} := \pi(q_i, \mathcal{T}), \quad y_i = \phi_{ground}(\mathcal{V}) \rightarrow \phi_{det}(\mathcal{V}) \rightarrow \phi_{qa}(\mathcal{V})$$

**CoTs Construction.** To ensure the correctness of outputs at intermediate steps, we leverage the training set from STAR for hyperparameter tuning, enabling us to identify the most effective model for each sub-task within the agent-based system. By following the provided programs, we evaluate the performance of the corresponding vision models on tasks such as object detection and action recognition. Given the availability of complete reasoning chains, we independently assess each sub-task using ground truth data for all preceding steps.

Table 1 presents the evaluation results for the various sub-tasks. For **question decomposition**, we compared several code LLMs, with DeepSeek-Coder-Instruct achieving the highest performance, outperforming even GPT-3.5-Turbo. In **object detection**, OWL-ViT v2 recorded the highest Intersection over Union (IoU) score, showcasing its superior open-vocabulary detection capability. The results for **temporal grounding** indicate that while UniVTG leads in performance, there remains a need for further advancements in this area. In **action recognition**, our evaluations showed that generative models outperformed discriminative models, likely due to the fine-grained action list provided by the STAR dataset. However, the performance of both model types reveals significant opportunities for improvement. Finally, in the **one-hop question answering** sub-task, all models

Table 1: Sub-tasks definition and evaluation results. We choose 3 model candidates for each sub-task and evaluate them in STAR training set with the corresponding metrics. Models with best performance are placed at the bottom of each column.

| Sub-task name | Model name | Metric | Number (%) |
|---|---|---|---|
| | CodeQwen1.5-Chat (7B) (Bai et al., 2023) | | 52.7 |
| Question decomposition | GPT-3.5-Turbo (OpenAI, 2023a) | Acc | 73.1 |
| | DeepSeek-Coder-Instruct (6.7B) (Daya et al., 2024) | | 85.7 |
| | OWL-ViT v1 (Matthias et al., 2022) | | 47.3 |
| Object detection | GLIP (Li* et al., 2022) | IoU | 58.9 |
| | OWL-ViT v2 (Minderer et al., 2024) | | 63.0 |
| | LITA (13B) (Huang et al., 2024) | | 11.7 / 20.2 |
| Temporal grounding | TimeChat (7B) (Ren et al., 2024) | IoU / Recall | 13.9 / 23.1 |
| | UniVTG (Lin et al., 2023) | | 24.7 / 35.3 |
| | InternVideo2 (1B) (Wang et al., 2024) | | 7.6 |
| Action recognition | Open-VCLIP (Weng et al., 2023) | Top1-Acc | 8.9 |
| | LLaVA-NeXT-Video-DPO (7B) (Zhang et al., 2024) | | 18.2 |
| | LLaMA-VID (7B) (Li et al., 2024c) | | 43.5 |
| Question answering | SeViLA (Yu et al., 2023a) | Acc | 46.5 |
| | LLaVA-NeXT-Video-DPO (7B) (Zhang et al., 2024) | | 53.4 |

performed admirably, with LLaVA-NeXT-Video-DPO standing out as a top performer, consistent with its strong results on other benchmarks.

With these high-performing models, we implement the agent-based approach on VideoQA datasets that consist solely of question-answer pairs. During the execution of the programs, we record all intermediate outputs to construct the CoTs. Since the outputs from these vision models vary in format—such as bounding boxes and free-form text—we employ another LLM to translate the execution trace into natural language, facilitating its use in the distillation process. Detailed examples are provided in Appendix B.

## 2.3 CoTs Verification

To further refine the quality of reasoning chains for VideoQA samples, we implement a two-step verification process: (i) we filter execution traces to retain only those where the program can reach correct output. For multiple-choice datasets, the output must match the correct answer exactly, while for open-ended datasets, we prompt the LLM to verify correctness, accounting for format differences; (ii) we prompt the LLM to evaluate the logical coherence and usefulness of the reasoning chains in solving the problem. The model assesses whether the CoTs follow a clear, step-by-step reasoning process and provides a binary evaluation ('Yes' or 'No') to indicate their quality (detailed prompts can be found in Appendix C). This two-step approach ensures that only accurate and high-quality CoTs are utilized for further distillation into the model.

After filtering, we provide the statistics for the generated CoTs on different datasets in Table 2. We primarily select compositional QA datasets, as these require the model to process spatial-temporal information from different events comprehensively.

## 2.4 Distill Step by Step

In this section, we describe the process of distilling the generated CoTs into a Video-LLM. This distillation enhances the model's ability for spatial-temporal video understanding and multi-step reasoning, thereby improving its performance on complex Video Question Answering (VideoQA) tasks.

Specifically, using the generated CoTs, we can build the dataset $D = \{(\mathcal{V}_j, q_j, \hat{y}_j, c_j, p_s)\}_{j=1}^N$, where $N$ is the total number of samples in the distilling dataset, $\mathcal{V}_j$ is the video input, $q_j$ is the

| Dataset | Description | # Labels | # CoTs |
|---------|-------------|----------|--------|
| AGQA | Compositional | 25.0K | 5.4K |
| ANetQA | Compositional | 25.0K | 3.6K |
| STAR | Compositional | 45.7K | 11.2K |
| NExT-QA | Temporal & Causal | 34.1K | 12.1K |
| CLEVRER | Spatial & Temporal | 21.0K | - |
| EgoQA | Ego-centric | 7.8K | - |
| **Total** | | **158.6K** | **32.3K** |

Table 2: Dataset statistics. The column "# Labels" indicates the number of VideoQA pairs, which include the video, query, possible answers (multiple-choice), and the correct answer. "# CoTs" refers to the number of CoTs generated using our agent-based system for each dataset.

question, $\hat{y}_j$ is the ground-truth answer, $c_j$ is the generated CoT, $p_s$ is the task-specific suffix prompt, to distinguish different tasks, for example, for multiple-choice VQA, the prompt is "Answer with the option's letter from the given choices directly and only give the best option", and for open-ended VQA, the prompt is "Answer in one word or phrase". Detailed prompts are provided in Appendix C.

At distillation stage, we minimize the cross-entropy loss of predicting both the answer and the chain-of-thoughts, we replace the suffix prompt $p_s$ with "Explain the rationale to answer the question" to control whether we want a question answer or a CoT to explain the thinking steps. Thus, our optimization objective is:

$$\mathcal{L} = \mathcal{L}_{\text{label}} + \lambda\mathcal{L}_{\text{rationale}} = \sum_{j=1}^{N} \ell(\Phi(\mathcal{V}_j, q_j, p_s), \hat{y}_j) + \lambda\ell(\Phi(\mathcal{V}_j, q_j, p_s), c_j)$$

Here we set $\lambda$ to 1 to ensure the importance of answer and rationale are equally considered, which can not only keep the capacity to predict the short question answer but also expand the ability to generate the rationale to solve the question. Notice that not all the QA pairs can generate qualified CoT. In that case, the $\mathcal{L}_{\text{rationale}}$ will be set to 0.

Table 3: Training and evaluation datasets statics.

| Dataset | Avg Duration (s) | Size train | Size eval | Type | Train | Eval |
|---------|------------------|------------|-----------|------|-------|------|
| **MC-VQA** | | | | | | |
| STAR (Wu et al., 2021) | 11.6 | 45.7K | 7.1K | Compositional | ✓ | ✓ |
| NExT-QA (Xiao et al., 2021) | 44 | 34.1K | 5.0K | Temporal & Causal | ✓ | ✓ |
| CLEVRER (Yi et al., 2020) | 5 | 21.0K | - | Spatial-temporal | ✓ | ✗ |
| Perception-Test (Pătrăucean et al., 2023) | 23 | - | 11.5K | General | ✗ | ✓ |
| MVBench (Li et al., 2024b) | 5-35 | - | 2.0K | General | ✗ | ✓ |
| VideoMME (Fu et al., 2024) | 1010 | - | 2.7K | General | ✗ | ✓ |
| **OE-VQA** | | | | | | |
| AGQA (Grunde-McLaughlin et al., 2021) | 30 | 25.0K | 2.0K | Compositional | ✓ | ✓ |
| ANetQA (Yu et al., 2023b) | 180 | 25.0K | 2.0K | Compositional | ✓ | ✓ |
| EgoQA (Grauman et al., 2022) | 6.4 | 7.8K | - | Ego-centric | ✓ | ✗ |
| Activitynet-QA (Yu et al., 2019) | 112 | - | 8.0K | General | ✗ | ✓ |
| Video-ChatGPT (Maaz et al., 2024) | 108 | - | 3.0K | General | ✗ | ✓ |

# 3 EXPERIMENTS

In this section, we present the experimental setup (Sec. 3.1) and comparison results on various VideoQA benchmarks (Sec. 3.2). Extensive ablation studies are also undertaken to further examine the contributions of our approach in Sec. 3.3, and an evaluation on the quality of rationales generated by the distilled model is made in Sec. 3.4.

Table 4: Comparison with Video-LLMs on MC-VQA benchmarks. LLaVA-NeXT-Video-AoTD outperforms all other baselines the and the version without CoT distillation.

| Model | MVBench (Acc.) | VideoMME (Acc.) | STAR (Acc.) | NExT-QA (Acc.) | Perception-Test (Acc.) |
|---|---|---|---|---|---|
| **Proprietary Models** | | | | | |
| Gemini 1.0 Pro (Google, 2023) | - | - | - | - | 51.1 |
| Gemini 1.0 Ultra (Google, 2023) | - | - | - | - | 54.7 |
| Gemini 1.5 Pro (Google, 2024) | - | 75.7 | - | - | - |
| GPT4-V (OpenAI, 2023b) | 43.7 | 60.7 | - | - | - |
| GPT4-O (OpenAI, 2024) | - | 66.2 | - | - | - |
| **Open-source Models** | | | | | |
| LLaMA-VID (7B) (Li et al., 2024c) | 41.9 | 25.9 | - | - | 44.6 |
| Video-LLaVA (7B) (Lin et al., 2024) | 41.0 | 40.4 | - | - | 44.3 |
| VideoChat2 (7B) (Li et al., 2024b) | 51.1 | 33.7 | 59.0 | 68.6 | 47.3 |
| VideoLLaMA2 (7B) (Cheng et al., 2024) | 53.4 | 44.0 | 58.5 | 62.3 | 49.6 |
| LLaVA-NeXT-Video (7B) (Zhang et al., 2024) | 46.5 | 41.0 | 52.4 | 61.6 | 47.5 |
| LLaVA-NeXT-Video-Instruct (7B) | 53.4 | 43.2 | 72.2 | 77.1 | 50.3 |
| LLaVA-NeXT-Video-AoTD (7B) | 55.6 | 45.0 | 74.3 | 77.6 | 50.6 |

## 3.1 Experimental Setup

**Base model.** We use LLaVA-NeXT-Video (7B) (Zhang et al., 2024) (LNV for short) as base Video-LLM, which has shown remarkable performance on image-centric tasks, for example image QA (Yue et al., 2024). We present comparison on naive instruction tuning with video question answering dataset or with additional CoTs distillation. For CoT conversion and verification, we prompt LLaMA-3.1-8B with the manually-designed instruction and some in-context examples. Detailed prompts are provided in Appendix C.

**Instruction tuning.** As shown in Table 2, we utilize both multiple-choice and open-ended QA data, along with the generated CoTs, to fine-tune the base video question answering model. The resulting distilled model is named **LLaVA-NeXT-Video-AoTD** (LNV-AoTD for short). Additionally, as baseline, we also train another version of the model using only the basic QA data, which we refer to as **LLaVA-NeXT-Video-Instruct** (LNV-Instruct for short).

**Evaluation benchmarks.** We conduct extensive evaluations on Multiple-Choice Video QA (MC-VQA) and Open-Ended Video QA (OE-VQA). We report the top-1 accuracy for all MC benchmarks, which means the proportion of the output equal to the answer. For the evaluation on AGQA and ANetQA, we sample subsets from them, due to the large volume of test set. We report a GPT-assessed accuracy and score with the help of GPT-3.5-turbo-0613 for all OE benchmarks. The accuracy is a binary right or wrong choice and the score means similarity of output to the answer. We evenly select the benchmark in-domain and out-of-domain for testing to ensure a comprehensive and reasonable evaluation of the model capability. Detailed statistics for evaluation benchmarks are shown in Table 3.

## 3.2 Quantitative Results

We divide the comparison into two parts: the first focuses on comparing the distilled model with other baselines, while the second examines the difference between the instruction-tuned model and the AoTD version. Note that, as the base model continues improving with more data and compute, we expect our proposed idea can be used to enhance the performance of any model.

**MC-VQA performance.** As shown in Table 4, our LLaVA-NeXT-Video-AoTD achieves superior performance across all benchmarks. Several key observations can be made: (i) Compared to the base model, even a simple instruction-tuning on certain VideoQA datasets significantly enhances the model's question-answering performance. This improvement is notable since the base model

Table 5: Comparison with Video-LLMs on OE-VQA benchmarks. LLaVA-NeXT-Video-AoTD improves performance in all open-ended benchmarks compared with the Instruct version.

| Model | ANetQA (Acc./Score) | AGQA (Acc./Score) | Video-ChatGPT (Score) | | | | | ActivityNet (Acc./Score) |
|---|---|---|---|---|---|---|---|---|
| | | | Corr. | Deta. | Cont. | Temp. | Cons. | |
| **Proprietary Models** | | | | | | | | |
| Gemini 1.0 Pro (Google, 2023) | - | - | - | - | - | - | - | 49.8/- |
| Gemini 1.0 Ultra (Google, 2023) | - | - | - | - | - | - | - | 52.2/- |
| Gemini 1.5 Pro (Google, 2024) | - | - | - | - | - | - | - | 56.7/- |
| GPT4-V (OpenAI, 2023b) | - | - | 4.09 | 3.88 | 4.37 | 3.94 | 4.02 | 59.5/- |
| GPT4-O (OpenAI, 2024) | - | - | - | - | - | - | - | 61.9/- |
| **Open-Source Models** | | | | | | | | |
| VideoLLaMA (7B) (Cheng et al., 2023) | - | - | 1.96 | 2.18 | 2.16 | 1.82 | 1.79 | 12.4/1.1 |
| Video-ChatGPT (7B) (Maaz et al., 2024) | - | - | 2.50 | 2.57 | 2.69 | 2.16 | 2.20 | 35.2/2.7 |
| LLaMA-VID (7B) (Li et al., 2024c) | - | - | 2.96 | 3.00 | 3.53 | 2.46 | 2.51 | 47.4/3.3 |
| Video-LLaVA (7B) (Lin et al., 2024) | - | - | 2.87 | 2.94 | 3.44 | 2.45 | 2.49 | 45.3/3.3 |
| VideoChat2 (7B) (Li et al., 2024b) | - | - | 3.02 | 2.88 | 3.51 | 2.66 | 2.81 | 49.1/3.3 |
| VideoLLaMA2 (7B) (Cheng et al., 2024) | - | - | 3.09 | 3.09 | 3.68 | 2.63 | 3.25 | 49.9/3.3 |
| LLaVA-NeXT-Video (7B) (Zhang et al., 2024) | 46.4/3.3 | 27.4/2.2 | 3.26 | 3.22 | 3.77 | 2.47 | 2.99 | 54.3/3.2 |
| LLaVA-NeXT-Video-Instruct (7B) | 47.1/3.1 | 59.3/3.4 | 2.96 | 2.81 | 3.35 | 2.42 | 2.82 | 50.0/3.3 |
| LLaVA-NeXT-Video-AoTD (7B) | 53.9/3.4 | 60.9/3.6 | 3.11 | 3.00 | 3.60 | 2.41 | 2.91 | 53.2/3.4 |

was primarily trained on static images and struggled with video understanding. (ii) Our model, trained with CoTs distillation, demonstrates further performance enhancements across all benchmarks, particularly on the compositional VideoQA benchmark (STAR) and comprehensive benchmarks (VideoMME, MVBench). This suggests that our AoTD method effectively improves the model's ability to address complex problems and interpret spatial-temporal scenes. (iii) The distilled model consistently outperforms all other baselines across all benchmarks, even when compared to more powerful models. This finding illustrates that our method effectively bridges performance gaps created by varying model components.

**OE-VQA performance.** As shown in Table 5, LLaVA-NeXT-Video-AoTD outperforms the Instruct variant across all open-ended VideoQA benchmarks. Notably, it achieves a greater percentage increase compared to the Multiple-Choice (MC-VQA) benchmarks, suggesting that CoTs distillation may be more effective for open-ended generation than for multiple-choice selection. While the distilled model scores higher than most models listed in the table, it does not surpass LLaVA-NeXT-Video on certain benchmarks. We conjecture this is due to the model's extensive training on images, that can also benefit the question answering without requiring complex reasonings, as also suggested by the findings in VideoLLaMA2 (Cheng et al., 2024). Additionally, the inherent challenges of evaluating open-ended VQA may influence the results. Assessments conducted by GPT can be biased or inaccurate, and the metrics we employ primarily indicate general trends rather than providing absolute accuracy.

### 3.3 ABLATION STUDY

**Analysis on CoT filtering.** To demonstrate the effectiveness of our filtering mechanism, we trained an alternative model without CoTs filtering while maintaining all other settings. The amount of CoTs distillation data increased to 36.3K. As shown in Table 6, the model's performance declines significantly on both the Multiple-Choice (MC-VQA) and Open-Ended VQA (OE-VQA) benchmarks when the CoT filtering mechanism is not utilized. This confirms that employing large language models (LLMs) to filter CoTs is an crucial for enhancing data quality.

**Analysis on model transferability.** As AoTD is a distillation method that leverages Chain-of-Thoughts (CoTs), it can theoretically be applied to any Video-LLMs. To assess the transferability of our method, we conduct experiments on another very recent model, LLaVA-OneVision(7B) (Li et al., 2024a). As shown in Table 6, our method still achieves significant improvements on the benchmarks, demonstrating both the transferability and robustness of the approach. Due to the rapid

advancements in the computer vision field, evaluating all models and benchmarks is prohibitively infeasible. Thus, we focus on assessing a single model against selected benchmarks to provide a representative evaluation.

## 3.4 EVALUATION ON RATIONALES

To verify whether the model has effectively learned multi-step reasoning through CoTs distillation, we analyze the rationales generated by the model. Specifically, we extract and evaluate the temporal and spatial information embedded within these rationales. This approach extends beyond merely assessing the correctness of the final answer, which could be influenced by biases or other external factors. By examining the reasoning process in detail, we gain a more accurate understanding of the model's ability to perceive and reason about spatial and temporal relationships.

**Evaluation protocols.** We randomly select 200 samples from the STAR validation set and perform inference on this subset using the suffix prompt, recording the generated rationales. From these rationales, we extract the predicted temporal windows and bounding boxes, comparing them to the ground truth. For the spatial evaluation, we calculate the IoU between the predicted and ground truth bounding boxes. For the temporal evaluation, we compute both IoU and Recall, leveraging the frame-level scene graph annotations provided in the STAR dataset.

**Evaluation results.** Table 7 presents the evaluation results. For comparison, we also test Uni-VTG for temporal reasoning and OWL-ViT v2 for spatial reasoning. The results show that LLaVA-NeXT-Video-Instruct struggles to generate valid rationales, even when using the suffix prompt. In contrast, LLaVA-NeXT-Video-AoTD demonstrates comparable performance to specialized models in both spatial and temporal reasoning, indicating that the model successfully acquired these abilities through the distillation process.

Table 6: Ablation results of CoT filtering and model transferability.

| Model | Filtering | MVBench (Acc.) | STAR (Acc.) | AGQA (Acc. / Score) |
|---|---|---|---|---|
| LNV-AoTD | ✗ | 53.7 | 73.3 | 59.5/3.5 |
| LNV-AoTD | ✓ | 55.6 | 74.3 | 60.9/3.6 |
| Onevision | - | 58.0 | 65.9 | 39.0/3.0 |
| Onevision-Instruct | - | 59.2 | 75.8 | 65.6/3.7 |
| Onevision-AoTD | ✓ | 60.5 | 76.6 | 65.7/3.7 |

Table 7: Temporal and spatial abilities evaluation result.

| Model | Temporal Grounding | | Spatial Grounding |
|---|---|---|---|
| | IoU (%) | Recall (%) | IoU (%) |
| UniVTG | 22.8 | 31.0 | - |
| OWL-ViT v2 | - | - | 64.7 |
| LNV-Instruct | ✗ | ✗ | ✗ |
| LNV-AoTD | 21.7 | 34.0 | 45.2 |

## 4 RELATED WORK

**Video-language models (Video-LLMs).** Most existing Video-LLMs are composed of a pre-trained visual encoder(like CLIP (Radford et al., 2021) or SigLIP (Zhai et al., 2023)) to encode video frames into a sequence of visual features, an adapter to transfer the visual features to tokens which can be understood by the language model, and a pre-trained LLM to output the final response. These models achieve strong ability for general vision-language tasks like Video question-answering (think the task as auto-regressive generation with question as prompt prefix). More recent works such as VideoLLaMA2 (Cheng et al., 2024), LLaVA-NeXT-Video (Zhang et al., 2024) and Videochat2 (Li et al., 2024b), with their excellent architecture design and reasonable instruction tuning data collection, have achieved a new level of zero-shot results in Video QA task. However, current end-to-end models still lack of interpretability for questions, as well as the ability to think and visually process complex problems in multiple steps, leads to their weakness in real complex scenarios, which is an important part for embodied learning and autonomous driving.

**Visual Programing and Agents.** With the progress of LLMs, some recent works (Gupta & Kembhavi, 2023; Surís et al., 2023) begin to try to use LLM as planner to solve the complex reasoning task in real scenarios. They attempt to decompose the question into some easier sub-questions, and use different specialist models as agents to solve these sub-questions, and finally gather them to get the answer of the raw question. MoReVQA (Min et al., 2024) proposes a multi-stage system, consisting

of an event parser, a grounding module, and a reasoning module with an external memory, getting a strong zero-shot Video QA ability while is able to create interpretable intermediate outputs. VURF (Mahmood et al., 2024) proposes a self-refinement method to resolve the LLM hallucinations to get a more concise program based on the context cues. These models demonstrate a strong ability to obtain trustworthy answers based on the intermediate evidence they get, but they lag far behind the end to end model in terms of inference speed, and often require some in-context examples to assist them in solving problems, which undoubtedly brings a lot of trouble to the use of these agent-based models.

**Chain-of-Thought (CoT).** Recent advancements in Chain-of-Thought (Wei et al., 2022; Yao et al., 2024) have made significant improvements in boosting the capabilities of LLMs. Though there have been several works enhancing the power of LLMs through distilling the CoT into the model, we still note a lack of research focused on applying CoT to video scenarios, as videos often have complex spatio-temporal relationships, and multi-step thinking is needed to solve the problems happened in these scenes. MotionEpic (Fei et al., 2024) develops a Video-of-Thought reasoning framework by integrating video spatial-temporal scene graph. But it requires explicit training on the graph encoder, which needs additional graph data, and cannot be directly migrated to other common Video-LLMs. Thus, we construct natural language CoTs which are involved with spatial-temporal information to adapt to any different models.

**Visual CoT.** The potential of Chain-of-Thought (CoT) reasoning extends beyond NLP to the visual domain. Several studies (Zhang et al., 2023; Mitra et al., 2024; Shao et al., 2024; Gao et al., 2024b) have applied CoT to visual understanding tasks, using powerful MLLMs for CoT generation or tool-based architectures for step-by-step problem solving. However, these methods face limitations, such as errors in CoT generation by MLLMs or high time and memory costs for tool-based approaches. Recent works like Visual Program Distillation (VPD) (Hu et al., 2024a) and Fact (Gao et al., 2024a) aim to maintain CoT accuracy and diversity while leveraging MLLMs to directly generate CoTs. These methods decompose complex tasks through code programs, invoking expert models to address sub-tasks, and use the generated CoTs as training data for fine-tuning visual-language models, thereby improving the model's ability to generate rationales directly. While all these methods focus on image-based domains, they overlook the video domain, where CoT is especially suitable due to the complex spatio-temporal nature of video understanding tasks. To bridge this gap, we propose AoTD, a method inspired by VPD and Fact, tailored to the video domain. Video-STaR (Zohar et al., 2024) also constructs CoTs using videos and existing labels for instruction tuning, without developing an agent-based system.

## 5 CONCLUSION & LIMITATION

In this work, we present Agent-of-Thought Distillation (AoTD), a novel approach aimed at distilling multi-step reasoning and spatial-temporal understanding into a large generative video-language model (Video-LLM). Our method introduces an agent-based system that automates the generation of Chain-of-Thoughts (CoTs) from various Video Question Answering (VideoQA) datasets by breaking down complex questions into manageable sub-tasks that can be addressed by specialized vision models. Extensive experiments validate that the distilled model significantly enhances performance on both Multiple-Choice (MC-VQA) and Open-Ended VQA (OE-VQA) benchmarks, underscoring the effectiveness of our approach.

Despite these advancements, several limitations remain and we leave them as future work: (i) Similar to prior approaches, the effectiveness of our agent-based system is contingent upon the progress of the underlying visual model components. Enhancing its ability to generalize across diverse datasets is essential for broader applicability. (ii) While our primary focus has been on compositional VideoQA tasks, and we have demonstrated improvements across a series of benchmarks, achieving holistic enhancements will require further exploration into creating a more balanced distribution of training data. (iii) Furthermore, our agent-based framework has the potential to address additional video-related tasks, such as video captioning and referring segmentation. We aim to expand our methodology to these domains, which could yield even more robust and versatile applications in the future. Overall, we believe AoTD represents a promising future direction for advancing multimodal reasoning abilities in Video-LLMs.

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

# A    EXPERIMENTAL DETAILS

## A.1    TRAINING DETAILS

For all models, their projection layers and language model are finetuned and visual encoder is frozen. We use a cosine learning rate schedule, with warm up ratio 0.03 and learning rate 4e-5. For both Instruct and AoTD setting, we finetune the model with batch size 48 and totally 1 epoch. We believe that longer training will get a better performance on in-domain benchmarks but maybe a destroy on out-of-domain benchmarks.

## A.2    SPECIALIZED MODELS EVALUATION DETAILS

In this section we will show the details about each sub-task's evaluation from data preparation to evaluation metric.

**Question Decomposition.**    Since there may be multiple valid ways to decompose the same problem, we evaluate only the accuracy of the final output in this sub-task. Specifically, the model takes the query and instruction as input and generates an executable program. We replace all intermediate outputs within the program and focus on whether the final output matches the correct answer. If the decomposition is correct, the final output must align with the answer. Any programs that cannot be executed or that lead to an incorrect answer are considered failures.

**Object Detection.**    To evaluate the performance of detection models, we sample frames with scene graph annotations from the input video clip and provide them, along with the text query, as input to the model. The model then outputs a series of bounding boxes that exceed a confidence threshold. We select the bounding box with the highest confidence as the final output and calculate the IoU to assess accuracy.

**Temporal Grounding.**    Since scene graphs provide both the start and end frame IDs, as well as key frame IDs for each event, we use IoU and Recall as metrics to capture different aspects of model performance. The model takes the video clip and text query as input and outputs the predicted start and end frame IDs. We calculate IoU based on the alignment between the predicted and annotated start and end frame IDs, and we compute Recall using the key frame ID annotations to evaluate how well the model captures important frames.

**Action Recognition.**    For discriminative models, we provide the video clip and a list of action labels as input to complete a classification task. For generative models, we provide the video clip along with an instruction prompt, asking the model to generate five actions most relevant to the video, ranked by likelihood. We then use the top-ranked output from each model to calculate the Top-1 accuracy for both approaches.

**Question Answering.**    The evaluation of question answering follows a similar approach to previous methods. The model takes the video clip and question as input and returns an answer, from which we directly calculate the accuracy. The key difference between this sub-task and a standard QA task is that the answers are based on a series of information collected by preceding agents, allowing for a more accurate assessment of the model's pure question-answering ability.

# B    MORE RESULTS

Here we introduce some examples to show the process from query to Chain-of-Thought using our agent-based pipeline. We can find that our pipeline is able to decompose complex questions into easier sub-tasks and the final CoT retains step-by-step problem-solving ideas and spatial-temporal information representing video understanding ability.

Question

Question: Why does the man have to throw the plane first in the middle of the video?
Possible answers: (A) chase the ball (B) the toy car hit the wall (C) put on earmuffs (D) get it start flying (E) bounce baby on cushion
Answer: (D) get it start flying

Program

```
def execute_command(video_clip, query, possible_answers:
    middle_clip = trim(video_clip, start=video_clip.num_frames//4, end=video_clip.num_frames//4*3)
    throw_clip = Filter_frames_with_act(middle_clip, 'man throw the plane')
    throw_captions = Video_caption(throw_clip)
    man_bboxes = Find(throw_clip, 'man')
    throw_reason = Video_query(throw_clip, query='why does the man have to throw the plane first',
                          possible_answers=possible_answers)
    info = {'reason why the man has to throw the plane first': throw_reason, 'what happened in the target
                                              video': throw_captions}
    answer = select_answer(query, info, possible_answers)
    return answer
```

Execution Trace

call function trim
trimmed video from frame 8 to frame 24
call Filter_frames_with_act
filter action man throw the plane
find action from frame 16 to frame 20
call function Video_caption
frame 16: A man and a child playing plane in a park.
frame 17: A man and a child standing in a grassy field.
frame 18: A man and a child standing in a grassy field.
frame 19: A man and a child playing plane in a park.
call function Find
finding man
find man at [283, 45, 330, 247] in frame 16
find man at [215, 117, 249, 263] in frame 17
find man at [189, 122, 220, 266] in frame 18
find man at [201, 132, 236, 277] in frame 19
call function Video_query
Question: why does the man have to throw the plane first
Answer: The man has to throw the plane first because it is a remote control airplane and needs to be launched into the air
        before the young boy can start playing with it.
call function select_answer
select: D
program output: get it start flying

CoT

To solve this question, we first need to find when the man is throwing the plane in the middle of the video. The man is throwing the plane from frame 16 to frame 20.
Next, we should find where the man is to better know why he has to throw the plane first. We can do this by analyzing the man's bounding box region in each frame. The man's bounding box is [283, 45, 330, 247] in frame 16, [215, 117, 249, 263] in frame 17, [189, 122, 220, 266] in frame 18, and [201, 132, 236, 277] in frame 19.
By analyzing the man's bounding box region, we can see that the man is playing with a remote control airplane and needs to launch it into the air before the young boy can start playing with it. This is because the airplane needs to be in the air for the boy to play with it.
Therefore, the answer is that the man has to throw the plane first because it is a remote control airplane and needs to be launched into the air before the young boy can start playing with it. So the answer should be get it start flying.

Figure 4: Example form NExT-QA (Xiao et al., 2021)

Figure 5: Example form ANetQA (Yu et al., 2023b)

## C    PROMPTS

In this section we present the prompts used in our agent-based pipeline for generating program, converting execution trace and filtering rationales.

### C.1    PROMPT FOR PROGRAM GENERATION

For each video and query, we call a LLM to decompose the query to a Python program under the guidance of the prompt below. We modify the ViperGPT (Surís et al., 2023) prompt to adapt to the visual agents we use.

```python
def Query_Objs(clip, query):
    """
    Query the objects that appear in video clip and match the query descriptions.
    Parameters
    -------
    clip:
        a list of video frames.
    query:
        Description of the target object.
    Returns
    -------
    a list of bounding boxes of the objects that match the query.
    Examples
    -------
    #return white_objs
    def execute_command(video_clip):
        white_objs = Query_Objs(video_clip, "white object")
        return white_objs
```

```python
    """

def Query_Actions(clip, obj=None):
    """
    Find the actions happened in the video clip, if obj is not None, query the actions related
 to it.
    Parameters
    -------
    clip:
        a list of the video frames.
    obj:
        object class which is used to query the actions related to it.
    Returns
    -------
    a list of actions classes happened in the video clip.
    Examples
    -------
    #return actions
    def execute_command(video_clip, query, possible_answers):
        actions = Query_Actions(video_clip)
        return actions
    """

def Filter_frames_with_act(clip, action):
    """
    filter a new video clip containing the time period in which the target action occurred
    Parameters
    -------
    clip:
        a list of video frames.
    action:
        the target action which is used to filter frames.
    Returns
    -------
    a new video clip ontaining the time period in which the target action occurred.
    Examples
    -------
    #return jump_clip
    def execute_command(video_clip, query, possible_answers):
        jump_clip = Filter_frames_with_act(video_clip, "person is jumping")
        return jump_clip
    """

def Filter_frames_with_obj(clip, obj):
    """
    filter a new video clip that the target object occured.
    Parameters
    -------
    clip:
        a list of video frames.
    obj:
        class or description about the target object.
    Returns
    -------
    a new video clip that the target object occured in it.
    Examples
    -------
    #return shoe_clip
    def execute_command(video_clip, query, possible_answers):
        shoe_clip = Filter_frames_with_obj(video_clip, "shoe")
        return shoe_clip
    """

def trim(clip, start=None, end=None):
    """
    Returns a new video clip containing a trimmed version of the original video at the [start,
 end] clip.
    Parameters
    ----------
    clip:
        a list of video frames.
    start : Union[int, None]
        An int describing the starting frame in this video clip with respect to the original
 video.
    end : Union[int, None]
        An int describing the ending frame in this video clip with respect to the original
 video.

    Returns
    -------
    a new video clip with start and end.
```

```python
    """
def Find(clip, obj):
    """
    find all bounding boxes around a certain object in the video clip,
    and collates them into a collection of frames.
    Parameters
    ----------
    clip:
        a list of video frames.
    obj:
        the object to look for.
    Returns
    -------
    a new video clip composed of crops of the object.
    Examples
    --------
    # Return the shoe_clip
    def execute_command(video_clip, query, possible_answers):
        shoe_clip = Find(video_clip, "shoe")
        return shoe_clip
    """

def select_answer(query, info, possible_answers):
    """
    Uses a language model to choose the option that best answers the question given the input
information.
    Parameters
    ----------
    query:
        the input question.
    info:
        Any useful information to answer the question.
    possible_answers:
        a list of possible answers to the question.
    Returns
    -------
    one answer chosen from the possible answers.
    Examples
    --------
    # Return the answer
    def execute_command(video_clip, query, possible_answers):
        clip_summary = Video_summary(video_clip)
        info = {
            "summary of the target video": clip_summary
        }
        answer = select_answer(query, info, possible_answers)
        return answer
    """
def exist(clip, query):
    """
    judge whether a object exists in the video.
    Parameters
    ----------
    clip:
        a list of video frames.
    query:
        query to the object class.
    Returns
    -------
    Return True if the object specified by query is found in the video, and False otherwise.
    Examples
    --------
    # Return the flag
    def execute_command(video_clip, query, possible_answers):
        flag = exist(video_clip, "shoe")
        return flag
    """
def Video_summary(clip, query):
    """
    give a brief summary of the video clip related to the query.
    Parameters
    ----------
    clip:
        a list of video frames.
    query:
        a question about the video.
    Returns
    -------
    return a brief summary of the video clip.
    Examples
    --------
```

```
176     # Return the clip_summary
177     def execute_command(video_clip, query, possible_answers):
178         clip_summary = Video_summary(video_clip, query)
179         return clip_summary
180     """
181 Write a function using Python and the functions (above) that could be executed to provide an
    answer to the query.
182
183 Consider the following guidelines:
184 - Use base Python (comparison, sorting) for basic logical operations, start/end, math, etc.
185 - Objects with mutiple names like "phone/camera", "cup/glass/bottle" with slash, input them as
     a whole object name.
186 - Just use the class and function appear above except for some base python operations.
187 - Only answer with a function starting def execute_command, do not answer any extra words and
    symbols before and after the function.
188 - No text that is not related to function can appear.
189 - the answer only begins with "def execute_command" and ends with "return answer".
190
191 Here are some examples of the function you should write:
192 -------
193 question: What else is the person able to do with the door?
194 possible answers: ["Hold the door.", "Put down the door.", "Close the door.", "Open the door."
    ]
195 def execute_command(video_clip, query, possible_answers):
196     door_clip = Filter_frames_with_obj(video_clip, "door")
197     person_clip = Find(door_clip, "person")
198     clip_summary = Video_summary(person_clip, query)
199     door_actions = Query_Actions(person_clip, "door", possible_answers=possible_answers)
200     door_actions =
201     info = {
202         "actions the person able to do with the door else": door_actions,
203         "summary of the target video": clip_summary
204     }
205     answer = select_answer(query, info, possible_answers)
206     return answer
207 -------
208 Query: INSERT_QUERY_HERE
209 possible answers: INSERT_POSSIBLE_ANSWERS_HERE
```

## C.2 PROMPT FOR EXECUTION TRACE CONVERSION

After getting the execution trace by running the program step by step, we use a LLM to convert the trace into a natural language CoT. The LLM takes query, execution trace, possible answers (in MC-VQA) and execution trace as input. The instruction prompt is as follow:

```
1 Given a video and a question, I wrote the function execute_command using Python, and the other
   functions above that could be executed to provide an answer to the query.
2 As shown in the code, the code will print execution traces.
3 I need you to rewrite the execution trace into a natural language rationale that leads to the
   answer.
4
5 Consider the following guidelines:
6 - Use all the bounding box information in the rationale, do not use words like "so on" to omit
   the bounding box, just write all of them into the rationale.
7 - Referencing the execution trace, write a reasoning chain that leads to the most common human
   answer. Notice that the output should be the same as the human answer, not necessarily the
   program output.
8 - If some part of the rationale lacks logic, add reasonable content to make it logical.
9
10
11 Here are some examples of the rantionale you should write:
12 -----
13 Question: What did the person do with the table?
14 def execute_command(video_clip, query, possible_answers, time_wait_between_lines, syntax):
15     table_clip = Filter_frames_with_act(video_clip, 'person interacting with table')
16     person_clip = Find(table_clip, 'person')
17     table_bboxes = Find(table_clip, 'table')
18     clip_summary = Video_summary(person_clip)
19     person_action = Query_Actions(person_clip, 'table', possible_answers=possible_answers)
20     info = {'actions the person do with the table': person_action, 'summary of the target
   video': clip_summary}
21     answer = select_answer(query, info, possible_answers)
22     return answer
23 Execution trace:
24 call Filter_frames_with_act
25 filter action person interacting with table
26 find action from frame 2 to frame 11
27 call function Find
```

```
28  finding person
29  find person at [139, 141, 229, 342] in frame 2
30  find person at [151, 123, 242, 349] in frame 3
31  find person at [153, 121, 242, 274] in frame 4
32  find person at [158, 123, 255, 261] in frame 5
33  find person at [163, 124, 270, 262] in frame 6
34  find person at [153, 121, 242, 351] in frame 7
35  find person at [95, 113, 196, 316] in frame 8
36  find person at [83, 113, 196, 285] in frame 9
37  find person at [112, 116, 201, 332] in frame 10
38  call function Find
39  finding table
40  find table at [183, 140, 269, 257] in frame 2
41  find table at [194, 131, 269, 255] in frame 3
42  find table at [227, 129, 269, 252] in frame 4
43  find table at [226, 165, 269, 258] in frame 5
44  find table at [233, 170, 270, 259] in frame 6
45  find table at [217, 129, 269, 256] in frame 7
46  find table at [217, 122, 270, 254] in frame 8
47  find table at [221, 123, 269, 256] in frame 9
48  find table at [225, 125, 270, 263] in frame 10
49  call function Video_summary
50  summary result: The video shows a man in a kitchen, bending over and holding an orange object,
     surrounded by various kitchen items and furniture, with a focus on his actions and the
     domestic setting.
51  call function Query_Actions
52  Query table
53  Answer: tidied up.
54  call function select_answer
55  the information used: - actions the person do with the table: tidied up.
56  - summary of the target video: The video shows a man in a kitchen, bending over and holding an
     orange object, surrounded by various kitchen items and furniture, with a focus on his actions
     and the domestic setting.
57  program output: Tidied up.
58  Rationale:
59  To solve this question, we first have to find when did the person interact with the table.
60  From the video, we can see that the person is interacting with the table from frame 2 to frame
     11.
61  In this time period, we can find person at [139, 141, 229, 342] in frame 2, [151, 123, 242,
     349] in frame 3, [153, 121, 242, 274] in frame 4 and so on.
62  Table can also be found at [183, 140, 269, 257] in frame 2, [194, 131, 269, 255] in frame 3,
     [227, 129, 269, 252] in frame 4 and so on.
63  By analyzing the person and table bounding box region, we can see that the person is holding
     an orange object to clean the table in the kirchen environment.
64  So the answer should be tidied up.
65  ------------------------------------------------
66  Now, look the question, program and execution trace, please transfer these information to a
     rantionale.
67  Question: INSERT_QUESTION_HERE
68  INSERT_PROGRAM_HERE
69  Execution trace:
70  INSERT_EXECUTION_TRACE_HERE
71  Rationale:
```

## C.3 PROMPT FOR CoT FILTERING

In order to obtain high quality distillation data, we continue using LLM to filter CoTs. We prompt the LLM to select those CoTs that are truly helpful for solving questions and reflect the step-by-stpe thinking process. The prompt is as follows:

```
1  I will give you a question and a rationale to solve the question, you need to judge whether
    the rationale is thinking step by step and helpful to solve the question.
2  If yes, return True, If not, return False. no need to explain.
3  Here is the question and rationale:
4  Question: INSERT_QUESTION_HERE
5  Rationale: INSERT_RATIONALE_HERE
```

## C.4 PROMPT FOR INFERENCE

Question: question content
Options:
(A) option content
(B) option content
(C) option content
(D) option content
Answer with the option's letter from the given choices directly and only give the best option. /
Explain the rationale to answer the question.

Question: question content
Answer in one word or phrase. / Explain the rationale to answer the question.

