# OpenReview forum: "Unlocking Video-LLM via Agent-of-Thoughts Distillation"
_ICLR.cc/2025/Conference — ICLR 2025 Conference Withdrawn Submission_

### Official Review · Reviewer_42Ti · 2024-10-21

**Soundness:** 2
**Presentation:** 3
**Contribution:** 1
**Rating:** 1
**Confidence:** 5

**Summary:**

AoTD introduced a method for converting VQA datasets into CoT-video datasets. This is done by utilizing different pre-trained VLMs (e.g., OWLv2, UniVTG, LLaVA-NeXT-Video-DPO) as ‘agents’ with frontier LLMs (e.g., chatGPT) to further supervise generation. They then showed that adding this ~158K dataset improved performance by ~1.5%.

**Strengths:**

- Training on the dataset did improve performance
- Nice integration of auxiliary models for data generation.
- I liked that open-source models were utilized rather than more expensive, proprietary

**Weaknesses:**

- Methodology very similar to “Visual Program Distillation: Distilling Tools and Programmatic Reasoning into Vision-Language Models”.
- Relatively small performance gain.
- Fails to compare/cite prior work, specifically (with reducing order of importance):
    * Visual Program Distillation - similar approach, using LLMs to generate programs and using agents to generate final answer
    * Video-STaR - also generates CoT video-sft data.
    * Visual CoT - CoT in images
    * Cantor : Inspiring Multimodal Chain-of-Thought of MLLM
    * Multimodal chain-of-thought reasoning in language models
    * Compositional chain-of-thought prompting for large multimodal models

**Questions:**

- Are any model’s trained during the data generation?
- Did you include any additional data during instruction tuning?

**Details Of Ethics Concerns:**

Upon further review, this paper is too similar to Visual Program Distillation (CVPR oral, 2024). I am especially suspicious of Fig. 2, which is very similar to VPD's figure 2. The method also appears almost identical, where:
1. An LLM is prompted to generate a 'program'
2. 'Agents' are called in a function-like format
3. An LLM is then used to re-progress the outputs into a final CoT response.
-> only applied to video.

However, the authors did not cite VPD. This combined with the visual similarity of Fig. 2 to the original Fig. 2 of VPD, leads me to flag this for ethical review.

VDP bibtex:
```bibtex
@InProceedings{Hu_2024_CVPR,
    author    = {Hu, Yushi and Stretcu, Otilia and Lu, Chun-Ta and Viswanathan, Krishnamurthy and Hata, Kenji and Luo, Enming and Krishna, Ranjay and Fuxman, Ariel},
    title     = {Visual Program Distillation: Distilling Tools and Programmatic Reasoning into Vision-Language Models},
    booktitle = {Proceedings of the IEEE/CVF Conference on Computer Vision and Pattern Recognition (CVPR)},
    month     = {June},
    year      = {2024},
    pages     = {9590-9601}
}
```

---

> ### Author Response · Authors · 2024-11-13
>
> Dear Reviewer,
>
> Thank you for pointing out the important citation oversight in our paper. We sincerely apologize for this mistake. During the final submission editing, we inadvertently deleted a paragraph that referenced two works that heavily inspired our research: VPD [1] and Fact [2]. We deeply regret this error and extend apologies to all reviewers, the authors of VPD and Fact for the oversight.
>
> That being said, we would like to clarify a few important points:
> 1. **Acknowledgment of reference:** As we said above, we acknowledge that this paper was inspired by VPD and Fact, so when drawing the architecture figure of our model (figure 2), we take their architecture figure for reference, but obviously there is a certain difference from them. For instance, we introduced an agent model selection process, which is not present in the works of VPD and Fact.
> 2. **No Plagiarism:** We want to emphasize that we there is **NO** plagiarism involved in this submission. The application of Chain-of-Thought (CoT) to large language models (LLMs) is a well-established research direction in NLP. While VPD effectively introduces CoT to the VLM domain, focusing primarily on images, we believe extending this approach to video was a natural progression. Our work builds on the transition from "language" to "image" to "video".
> 3. **Key Differences in Contribution:** Beyond the shared inspiration, our work introduces two main contributions that differ from VPD:
>    - **Agent-based system**: We use the STAR [3] dataset to systematically evaluate various expert model candidates in real-world QA scenarios, and select the best performers to form our agent-based system, rather than simply filtering models based on their performance in proprietary datasets in their domain. This ensures the generalization and overall performance of the agent-based system in real complex scenarios, and provides a guarantee for the quality of CoT data.
>     - **Quantitative Analysis**: While VPD only performs qualitative analysis (human evaluation) of the rationales generated by the distilled model, we have conducted a more rigorous quantitative analysis, assessing the generated rationales in terms of both time and space. Additionally, we compare our distilled model's performance with expert models, demonstrating that our model has successfully learned key capabilities from the experts.
> 7. **Correction of Oversight:** We will promptly restore the missing references to VPD and Fact and upload a revised version of the paper. Once again, we deeply apologize for this oversight and the resulting confusion.
>
> Given the clarifications above, we kindly the reviewer to reconsider the rating on our paper, given the citation issue has been addressed. Every submission opportunity is incredibly valuable to us, and we hope this mistake does not unduly affect the outcome of our work.
>
> Due to the importance of the missing citation issue, we have prioritized addressing your related concerns first. We will respond to your other questions at a later time.
>
> Thank you again for your understanding, and please accept our sincerest apologies.
>
> [1] Hu et al. Visual Program Distillation: Distilling Tools and Programmatic Reasoning into Vision-language Models. https://arxiv.org/abs/2312.03052. CVPR.
>
> [2] Gao et al. Fact: Teaching MLLMs with Faithful, Concise and Transferable Rationales. https://arxiv.org/abs/2404.11129. ACM MM.
>
> [3] Wu et al. STAR: A Benchmark for Situated Reasoning in Real-World Videos. https://arxiv.org/abs/2405.09711. NeurIPS.

---

### Official Review · Reviewer_tAzy · 2024-11-05

**Soundness:** 2
**Presentation:** 3
**Contribution:** 2
**Rating:** 5
**Confidence:** 3

**Summary:**

The authors propose Agent-of-Thoughts Distillation (AoTD), a method that uses an agent-based system to decompose complex video questions into manageable sub-tasks, each handled by a specialized model. The system generates Chain-of-Thoughts (CoTs) reasoning paths and incorporates a verification mechanism using LLMs to ensure reliability. These verified reasoning paths are then distilled into a single video-language model during instruction-tuning.

**Strengths:**

-	The approach of combining agent-based systems with chain-of-thought distillation to develop an end-to-end model is well-motivated (faster inference speed, reduced memory needed to save pre-trained models like in visual programming approach).
-	The experiments are fairly comprehensive.

**Weaknesses:**

•	Although the method shows potential, several experimental setups require further clarification (see the question section below).
•	The method heavily relies on LLMs for both program-to-CoT conversion and filtering, which may explain the limited improvement of qualified Chain-of-Thoughts rationales (only 20% of original QA pairs generated qualified CoTs - 32.3K out of 158.6K).

**Questions:**

Regarding the experiment results:

-	The method employs multiple component submodels, which likely will fuel up computational costs during training. However, the performance improvements over baselines appear insignificant: Table 4 shows a maximum gain of only 2.2% compared to standard instruction tuning. Table 5 even shows that the method performs worse than the base LLaVA-NeXT-Video model, suggesting the proposed fine-tuning approach might actually hurt the performance in some cases.

-	The study uses LLaVA-NeXT-Video-DPO as an expert for question-answering distillation but lacks direct performance comparisons. Given that the proposed model can generate rationales, it would be valuable to evaluate whether it outperforms LLaVA-NeXT-Video-DPO on more complex questions.

- The paper would benefit from analyzing model performance across different levels of question complexity. This analysis could better demonstrate the method's effectiveness in rationale generation.

- Has there been any evaluation of the LLM's performance in converting code to CoTs and filtering them? Would using a more capable LLM result in more qualified CoTs and subsequently improve model performance? What is the relationship between CoT quality and the final model performance?

---

### Official Review · Reviewer_jfoX · 2024-11-06

**Soundness:** 2
**Presentation:** 3
**Contribution:** 2
**Rating:** 6
**Confidence:** 3

**Summary:**

This paper contributes a stepwise reasoning pipeline for VQA task, namely Agent-of-Thoughts Distillation (AoTD). AoTD decomposes complex questions into manageable sub-tasks and leverages selected agent visual models to solve them sequentially. The execution traces are translated to the natural language CoTs to enhance the instruction-tuning for large generative Video-LLMs. The authors also design a verification method using a LLM to assess the quality of generated CoTs.

**Strengths:**

1. The paper is well-structured and clearly written, making it easy to follow. The contextualized related work highlights the relevance and timeliness of this research.

2. AoTD is a straightforward yet brilliant mechanism for improving the explainability and spatial-temporal understanding of the Video-LLMs in VQA task, with potential transferability in other tasks such as referring segmentation.

3. The experiments demonstrate the effectiveness of AoTD, where it achieves the SOTA performance compared to the selected open-source baselines in all multiple-choice VQA benchmarks and partial open-ended VQA benchmarks (ANetQA & AGQA).

**Weaknesses:**

1. A weakness that can be ignored is that there are still some typos.

2. AoTD is only applied to the LLaVA-NeXT-Video in experiments. To strengthen the study, it would be valuable to verify the generalizability of AoTD for other compared models, such as the VideoLLaMA2.

3. The work lacks experiments exploring the impact of different values for the $\lambda$ in $L=L_\text{label}+\lambda L_{\text{rationale}}$, which would be helpful to quantify the importance of rationale for question solving.

4. Please report the inference time required by the LLaVA-NeXT-Video-AoTD in comparison with the baselines, as this information is crucial for understanding the practicality of AoTD.

**Questions:**

A primary question pertains to the results reported in Tables 4 and 5. It appears that some values were taken from the VideoLLaMA2 and LLaVA-NeXT-Video papers. However, the results for the LLaVA-NeXT-Video in Table 5 are lower than those in Table 6 of the VideoLLaMA2 paper, despite other baseline results remaining consistent as reported in the VideoLLaMA2 paper. Could the authors explain this?

The MVBench result for the VideoLLaMA2 in Table 4 is the same as reported in the VideoLLaMA2 paper with an 8-frame setting. However, the VideoMME result for the VideoLLaMA2 in Table 4 is lower than the VideoLLaMA2 paper in the same setting, which is actually higher than the LLaVA-NeXT-Video-AoTD. Furthermore, the Perception-Test result reported in the VideoLLaMA2 paper with a 16-frame setting outperforms the LLaVA-NeXT-Video-AoTD. If the LLaVA-NeXT-Video-AoTD operates with 32 frames, it raises concerns about the fairness.

Please respond to the concerns listed in Weaknesses and Questions. I will raise my score if they are fairly discussed or addressed.

---

### Official Review · Reviewer_1h3R · 2024-11-07

**Soundness:** 3
**Presentation:** 3
**Contribution:** 3
**Rating:** 6
**Confidence:** 4

**Summary:**

This paper focuses on video question answering and proposes an instruction-tuning method called Agent-of-Thoughts Distillation (AoTD) for Video-LLMs. The authors first employ an agentic system to decompose complex video questions into sub-tasks which are iteratively solved by cutting-edge visual models to finally answer the video questions. The execution traces of this agentic system are then converted into chain-of-thoughts (CoTs), which are used for tuning Video-LLMs to generate answers with clear reasons and explainability. The authors conduct extensive experiments to show the effectiveness of AoTD in getting more accurate results and rationales on both multiple-choice video questions and open-ended video questions.

**Strengths:**

1. The paper proposes a novel attempt to use CoTs to finetune video-LLMs for generating better answers with reasons, which is different from agent tuning where CoTs are traditionally used for finetuning LLMs for generating better planning and tool calls.
2. The motivation of the paper is clear: using CoTs to finetune Video-LLMs in order to enhance the reasoning, spatial grounding and temporal grounding ability of the end-to-end models. The paper is well written.
3. Extensive experiments and ablation study are conducted to illustrate the effectiveness of AoTD in boosting performance as well as endowing Video-LLMs with multi-step reasoning, temporal grounding and spatial grounding abilities.

**Weaknesses:**

1. The paper does not provide qualitative examples of the answers and rationales of the Video-LLMs tuned by AoTD.
2. The generalization ability of AoTD on diverse question types remains to be discussed, since not all the video questions can be decomposed into the sub-tasks. For example, “explain the humor in this video”.
3. Only perform object detection on video frames is not sufficient enough to accurately understand the video. Object tracking and object re-identification should be taken into consideration for temporal consistency.

**Questions:**

1. What is the average length of the tested videos? Since the Video-LLMs receives a fixed number of frames, what if the key frames are not sampled for the model input because of the long video length? What will the rationale output be like in these cases?
2. Will hallucination happen in the generated rationale? If so, how to avoid hallucination?
3. In line 200, It seems that the CoTs of the agentic system follows a fixed order of tool calls: temporal grounding, object detection, and then question answering. Why using a fixed tool-call sequence instead of letting the LLM agent to decide the tool call sequence by its own?
4. How will the tuned Video-LLMs answer the question like: “what is the main idea of the video?”

---

### Official Review · Reviewer_Y5kG · 2024-11-12

**Soundness:** 3
**Presentation:** 3
**Contribution:** 2
**Rating:** 5
**Confidence:** 2

**Summary:**

This paper proposes a method to enhance large generative video-language models (Video-LLMs) for Video Question Answering (VideoQA) by incorporating Chain-of-Thoughts (CoTs) into their instruction-tuning process, improving both their accuracy and interpretability. VideoQA is an essential task that allows AI systems to respond to visual questions in videos, which requires understanding actions, objects, and scenes dynamically. Traditional Video-LLMs often lack explainability and struggle with their answers in spatio-temporal details, while agent-based systems offer greater interpretability but are limited by high resource demands and the current limitations of video understanding tools. To address these issues, the authors introduce a strategy that automatically generates CoTs by sub-tasks, each managed by specialized vision models. They evaluate these models on fundamental tasks like action recognition to identify the tools for each sub-task. The CoTs are then verified for logical consistency by a large language model, ensuring that only high-quality reasoning paths are distilled into the Video-LLM. This combined approach enhances the model’s ability to provide accurate answers along with interpretable reasoning, performing better on VideoQA benchmarks.

**Strengths:**

1. The approach exploits the benefits on the top of the insightful methods and effectively breaks down complex tasks into simpler sub-tasks, enabling more interpretable inference. This strategy enhances the model’s ability to handle intricate video content by focusing on multiple types of tasks.
2.  The use of CoTs and verification policy aligns the model’s inference process, making it easier to understand how answers are derived. This supports better performance in benchmarks that require diverse video content.

**Weaknesses:**

1. While the paper introduces the use of Chain-of-Thoughts (CoTs) to enhance Video-LLMs and improve interpretability, the approach mainly builds on existing CoT, that have been explored in other question-answering contexts. The integration of CoTs and verification through a large language model adds value, but the passrate and clarity verification concepts are explored in the existing work and may be seen as an extension rather than a novel contribution in the method.
2. Despite improvements shown in the multiple-choice VideoQA benchmarks, the method’s performance gains in open-ended VQA tasks are more modest. Open-ended questions require more flexible and nuanced interpretations that challenge even advanced models with sophisticated processes. The method dependency on pre-defined sub-tasks and existing visual understanding tools may restrict its adaptability and performance when faced with the variability and depth required for open-ended video-based questions.
3. While the method of subtasks and employing specialized vision tools for handling sub-tasks contributes to improved interpretability, it also raises questions regarding efficiency. The use of multiple tools, each responsible for distinct parts of the video analysis (e.g., action recognition, temporal segmentation), can lead to higher computational costs and longer processing times. This multi-step approach may hinder real-time video processing. Therefore, its deserve to analyze the multiple external tools in practical efficiency, and conclude if its suitable.
4. Another notable weakness is that the code or relevant materials (such as prompts) for the proposed method have not yet been attached. This limitation restricts for reviews, which is important for advancing the field too. Open-source availability is particularly important in areas like VideoQA, where benchmark comparisons and collaborative improvements drive progress. This restricts the transparency and broader adoption of the approach.

**Questions:**

Refer to above questions

**Details Of Ethics Concerns:**

How to avoid the bias or ethic outputs for the video LLMs

---

### Note · Authors · 2024-11-14

**Comment:**

Thanks to all the reviewers for their feedbacks, we will continue to improve our work!

**Withdrawal Confirmation:**

I have read and agree with the venue's withdrawal policy on behalf of myself and my co-authors.